# AutoOED: Automated Optimal Experimental Design Platform with Data- and Time-Efficient Multi-Objective Optimization

## Abstract

We present AutoOED, an Automated Optimal Experimental Design platform powered by machine learning to accelerate discovering solutions with optimal objective trade-offs. To solve expensive multi-objective problems in a data-efficient manner, we implement popular multi-objective Bayesian optimization (MOBO) algorithms with state-of-the-art performance in a modular framework. To further accelerate the optimization in a time-efficient manner, we propose a novel strategy called Believer-Penalizer (BP), which allows batch experiments to be accelerated asynchronously without affecting performance. AutoOED serves as a testbed for machine learning researchers to quickly develop and evaluate their own MOBO algorithms. We also provide a graphical user interface (GUI) for users with little or no experience with coding, machine learning, or optimization to visualize and guide the experiment design intuitively. Finally, we demonstrate that AutoOED can control and guide real-world hardware experiments in a fully automated way without human intervention.

## 1 Introduction

Optimal Experimental Design (OED) problems in science and engineering often require satisfying several conflicting objectives simultaneously. These problems aim to solve a multi-objective optimization system and discover a set of optimal solutions, called Pareto optimal. Furthermore, the objectives are typically black-box functions whose evaluations are time-consuming and costly (e.g., measuring real experiments or running expensive numerical simulations). Thus, the budget that determines the number of experiments can be heavily constrained. Hence, an efficient strategy for guiding the experimental design towards Pareto optimal solutions is necessary. Recent advances in machine learning have facilitated optimization of various design problems, including chemical design (Griffiths & Hernández-Lobato, 2017), material design (Zhang et al., 2020), resource allocation (Wu et al., 2013), environmental monitoring (Marchant & Ramos, 2012), recommender systems (Chapelle & Li, 2011) and robotics (Martinez-Cantin et al., 2009). A machine learning concept that enables automatic guidance of the design process is Bayesian optimization (Shahriari et al., 2016). This concept is extensively studied in the machine learning community from a theoretical aspect and in the single-objective case. However, its practical applications in multi-objective problems are still not widely explored due to the lack of easy-to-use and open-source software.

In this paper, we present AutoOED[1], an open-source platform for efficiently optimizing multi-objective problems with a restricted budget of experiments. The key features of AutoOED include:

- **Data-efficient experimentation**: AutoOED employs state-of-the-art MOBO strategies that rapidly advances the Pareto front with a small set of evaluated experiments.
- **Time-efficient experimentation**: AutoOED supports both synchronous and asynchronous batch optimization to accelerate the optimization. We propose a novel and robust asynchronous optimization strategy named Believer-Penalizer (BP), which is instrumental when multiple workers run experiments in parallel, but their evaluations drastically vary in time.

---

[1]Code, screenshots, detailed documentation and tutorials can be found at `https://sites.google.com/view/autooed`.

- **Intuitive GUI**: An easy-to-use graphical user interface (GUI) is provided to directly visualize and guide the optimization progress and facilitate the operation for users with little or no experience with coding, optimization, or machine learning.

- **Modular structure**: A highly modular Python codebase enables easy extensions and replacements of MOBO algorithm components. AutoOED can serve as a testbed for machine learning researchers to easily develop and evaluate their own MOBO algorithms.

- **Automation of experimental design**: The platform is designed for straightforward integration into a fully automated experimental design pipeline as long as the experiment evaluations (either in simulation or physical) can be controlled via computer programs.

## 2 RELATED WORK

**Bayesian optimal experimental design** Optimal experimental design (OED) is the process of designing the sequence of experiments to maximize specific objectives in a data- or time-efficient manner. Therefore, Bayesian optimization (BO) (Shahriari et al., 2016) is usually applied to find optimal solutions with a minimal number of evaluations. Essentially, BO relies on *surrogate models* like the Gaussian process (GP) to accurately model the experimental process and proposes new experimental designs based on defined *acquisition functions* that trade-off between exploration and exploitation. Popular choices of the acquisition functions include Expected Improvement (EI) (Močkus, 1975), Upper Confidence Bound (UCB) (Srinivas et al., 2010), Thompson Sampling (TS) (Thompson, 1933). Bayesian OED has found success in a wide range of applications (Greenhill et al., 2020) and is the main methodology of AutoOED. To further speed up when evaluations can be carried out in parallel, asynchronous BO approaches have been developed (Ginsbourger et al., 2010; Kandasamy et al., 2018; Alvi et al., 2019). However, all of the previous literature focuses on single-objective BO rather than the multi-objective scenario. In this paper, we extend several single-objective asynchronous BO methods to multi-objective versions and propose a novel asynchronous method named Believer-Penalizer (BP) with the stablest performance on multi-objective benchmark problems.

**Multi-objective Bayesian optimization** MOBO is developed to optimize for a set of Pareto optimal solutions while minimizing the number of experimental evaluations. Early approaches solve multi-objective problems by scalarizing them into single-objective ones using random weights (Knowles, 2006). Instead of scalarization, some acquisition functions are proposed to compute a single objective, e.g., entropy-based or hypervolume-based (Russo & Van Roy, 2014; Belakaria et al., 2019; Emmerich & Klinkenberg, 2008; Daulton et al., 2020a). Alternatively, MOBO can be solved by defining a separate acquisition function per objective, optimizing using cheap multi-objective solvers (usually evolutionary algorithms like NSGA-II (Deb et al., 2002)) and finally selecting one or a batch of designs to evaluate next (Bradford et al., 2018; Belakaria & Deshwal, 2020; Konakovic Lukovic et al., 2020). AutoOED implements many of them in a modular way and allows easily changing modules in an unified MOBO framework (see Section 3.2).

**Open-source Bayesian optimization platform** There are many existing Python libraries for Bayesian optimization including Spearmint (Snoek et al., 2012), HyperOpt (Bergstra et al., 2013), GPyOpt (authors, 2016), GPflowOpt (Knudde et al., 2017), Dragonfly (Kandasamy et al., 2020), AX (Bakshy et al., 2018), Optuna (Akiba et al., 2019), HyperMapper (Nardi et al., 2019), BoTorch (Balandat et al., 2020a), SMAC3 (Lindauer et al., 2021) and OpenBox (Li et al., 2021). These Python libraries are designed for general applications and have different algorithmic features supported. The feature comparison between AutoOED and these libraries is shown in Table 1 and is further discussed in Section 5.2. However, they are all targeted for experts in coding without an intuitive GUI. In contrast, there are also software platforms that provide intuitive user interface and visualization to specific domain experts but the platforms cannot be used for other general applications, for example, Auto-QChem (Shields et al., 2021) for chemical synthesis and GeoBO (Haan, 2021) for geoscience. Combining powerful Bayesian optimization algorithms and an intuitive GUI, AutoOED is designed to be a general optimization platform that can be easily used by anyone for applications in any field.

---

[2]The comparison is based on AutoOED's core features. "∼" means the package only supports a single multi-objective algorithm rather than a modular multi-objective framework with several state-of-the-art algorithms.

Table 1: Feature comparison between different Bayesian optimization platforms.[2]

| Name | GUI | Multiple objectives | Multiple domains | Asynchronous optimization | External evaluation | Modular framework | Built-in visualization |
|---|---|---|---|---|---|---|---|
| Spearmint | | | ✓ | | | | |
| GPyOpt | | | ✓ | | ✓ | ✓ | ✓ |
| GPflowOpt | | ∼ | | | | ✓ | |
| Dragonfly | | ∼ | ✓ | ✓ | | | |
| BoTorch | | ✓ | | ✓ | | ✓ | |
| **AutoOED** | ✓ | ✓ | ✓ | ✓ | ✓ | ✓ | ✓ |

## 3 DATA-EFFICIENT MULTI-OBJECTIVE OPTIMIZATION

### 3.1 PROBLEM FORMULATION

Optimal experiment design problems involving multiple conflicting objectives can be formulated as a multi-objective optimization on design parameters as data- and time-efficient as possible. More formally, we consider a optimization problem over a set of design variables $\mathcal{X} \subset \mathbb{R}^d$, called *design space*. The goal is to simultaneously minimize $m \geq 2$ ob-

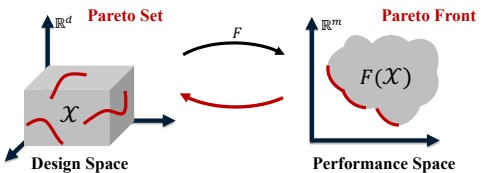

jective functions $f_1, ..., f_m : \mathcal{X} \rightarrow \mathbb{R}$. Representing the vector of all objectives as $\mathbf{f}(\mathbf{x}) = (f_1(\mathbf{x}), ..., f_m(\mathbf{x}))$, the *performance space* is then an $m$-dimensional image $\mathbf{f}(\mathcal{X}) \subset \mathbb{R}^m$. Conflicting objectives result in a set of optimal solutions rather than a single best solution. These optimal solutions are referred to as *Pareto set* $\mathcal{P}_s \subseteq \mathcal{X}$ in the design space, and the corresponding images in performance space are *Pareto front* $\mathcal{P}_f = \mathbf{f}(\mathcal{P}_s) \subset \mathbb{R}^m$.

To measure the quality of an approximated Pareto front, *hypervolume* (Zitzler & Thiele, 1999) is the most commonly used metric in multi-objective optimization (Riquelme et al., 2015). Let $\mathcal{P}_f$ be a Pareto front approximation in an $m$-dimensional performance space and given a reference point $\mathbf{r} \in \mathbb{R}^m$, the hypervolume $\mathcal{H}(\mathcal{P}_f)$ is defined as $\mathcal{H}(\mathcal{P}_f) = \int_{\mathbb{R}^m} \mathbb{1}_{H(\mathcal{P}_f)}(z)dz$, where $H(\mathcal{P}_f) = \{z \in Z \mid \exists 1 \leq i \leq |\mathcal{P}_f| : \mathbf{r} \preceq z \preceq \mathcal{P}_f(i)\}$. $\preceq$ is the relation operator of objective dominance and $\mathbb{1}_{H(\mathcal{P}_f)}$ is a Dirac delta function that equals 1 if $z \in H(P_f)$ and 0 otherwise. The higher the hypervolume, the better $\mathcal{P}_f$ approximates the true Pareto front.

### 3.2 MODULAR ALGORITHM FRAMEWORK

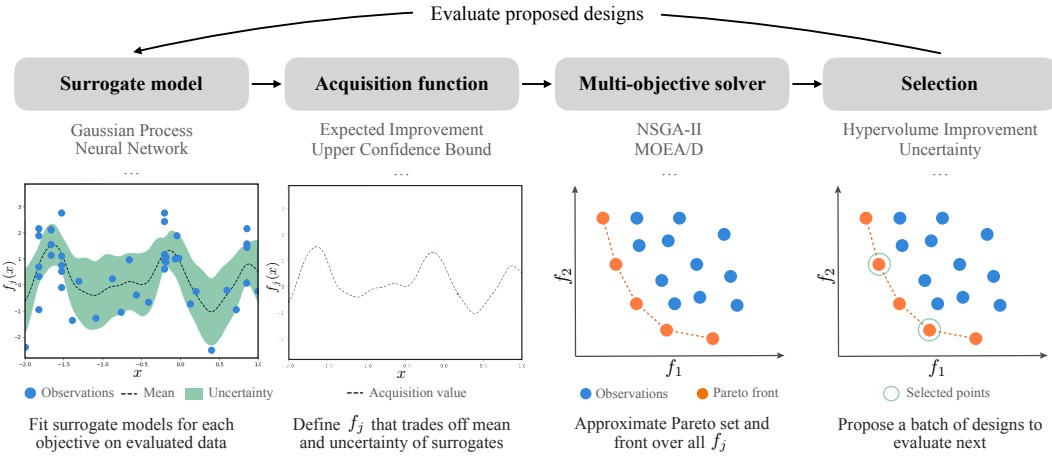

Figure 1: Algorithmic pipeline and core modules of multi-objective Bayesian optimization.

Multi-objective Bayesian optimization (MOBO) is a data-driven approach that attempts to learn the black-box objective functions $\mathbf{f}(\mathbf{x})$ from available data and find Pareto optimal solutions in an iterative and data-efficient manner. MOBO typically consists of four core modules: (*i*) an inexpensive *surrogate model* for the black-box objective functions; (*ii*) an *acquisition function* that defines sampling from the surrogate model and trade-off between exploration and exploitation of the design space; (*iii*) a cheap *multi-objective solver* to approximate the Pareto set and front; (*iv*) a *selection strategy* that proposes a single or a batch of experiments to evaluate next. These four modules (see Figure 1) are implemented as core and independent building blocks of the AutoOEDs, making it highly modularized and easy to develop new algorithms and modules. The whole pipeline starts from a given small dataset or a set of random evaluated samples, then it works iteratively by proposing new design samples and evaluating them until the *stopping criterion* is met.

For each module in this framework, AutoOED supports following choices:

- **Surrogate model**: Gaussian process, neural network (multi-layer perceptron), Bayesian neural network (DNGO (Snoek et al., 2015))
- **Acquisition function**: Expected Improvement, Probability of Improvement, Upper Confidence Bound, Thompson Sampling, identity function
- **Multi-objective solver**: NSGA-II, MOEA/D, ParetoFrontDiscovery (Schulz et al., 2018)
- **Selection**: Hypervolume improvement, uncertainty, random, etc.
- **Stopping criterion**: Time, number of evaluations, hypervolume convergence

```
class TSEMO(MOBO):            class USEMO_EI(MOBO):         class DGEMO(MOBO):
'''                           '''                           '''
[Bradford et al. 2018]        [Belakaria and Deshwal 2020]  [Lukovic et al. 2020]
'''                           '''                           '''
spec = {                      spec = {                      spec = {
    'surrogate': 'gp',            'surrogate': 'gp',            'surrogate': 'gp',
    'acquisition': 'ts',          'acquisition': 'ei',          'acquisition': 'identity',
    'solver': 'nsga2',            'solver': 'nsga2',            'solver': 'discovery',
    'selection': 'hvi',           'selection': 'uncertainty',   'selection': 'direct',
}                             }                             }
```

Code Example 1: Creating algorithms in AutoOED by simply specifying module combinations.

Based on this framework, we implement several popular and state-of-the-art MOBO methods, including ParEGO (Knowles, 2006), MOEA/D-EGO (Zhang et al., 2009), TSEMO (Bradford et al., 2018), USeMO (Belakaria & Deshwal, 2020), DGEMO (Konakovic Lukovic et al., 2020). DGEMO exhibits state-of-the-art performance for data-efficient, multi-objective problems with batch evaluations to the best of our knowledge. With necessary modules of the MOBO framework implemented, the algorithms can be easily composed by specifying the choice of each module and inheriting the base class MOBO, see Code Example 1. Supported choices of each module can be found in our documentation. Users can select an algorithm from this library that best fits the characteristics of their physical system or optimization goals, or they can easily create new algorithms by specifying novel combinations of existing modules in just a few lines of code.

## 4 TIME-EFFICIENT MULTI-OBJECTIVE OPTIMIZATION

### 4.1 BATCH OPTIMIZATION

While standard MOBO optimizes for the Pareto front in a data-efficient manner, often, when multiple experiment setups are available, evaluations can be executed in batch by parallel workers to further speed up the whole optimization process. To leverage this speed-up, all the algorithms in AutoOED are implemented to support batch evaluation.

However, if parallel workers evaluate in different speeds, some workers are left idle when they finish evaluations earlier than others. Therefore, asynchronous optimization is desired to maximize the utilization of workers and is able to evaluate many more designs than synchronous optimization in a fixed amount of time, as also illustrated by Kandasamy et al. (2018) and Alvi et al. (2019). Nevertheless, while some designs are being evaluated (i.e., busy designs), how to propose the next

design that *(i)* avoids being similar to the busy designs and *(ii)* incorporates knowledge from busy designs to reach better regions in the performance space is the key question that we want to explore.

To develop efficient asynchronous strategy for multi-objective optimization, we borrow ideas from previous literature in the single-objective setting.

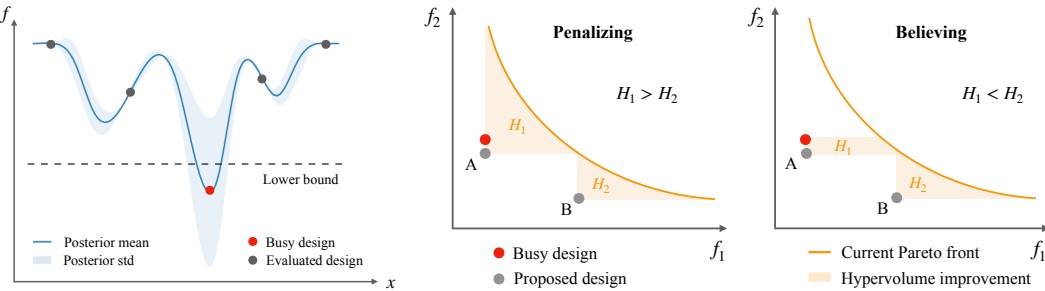

(a) The failure case of KB when believing overestimated busy designs.

(b) The sub-optimality of LP in multi-objective scenario when believing busy designs affects the selection result.

Figure 2: Analysis of KB and LP strategies for asynchronous optimization.

### 4.2 FAILURES OF EXISTING STRATEGIES

**Kriging Believer** (KB) (Ginsbourger et al., 2010) is a simple yet effective approach that believes the performance of busy designs is their posterior mean of the surrogate model when optimizing for new designs. In other words, it treats the mean prediction of the busy designs as their real performance and eliminates their posterior variance to prevent acquisition functions from preferring those regions. However, failure case happens when it believes an overestimated design, it might become difficult to find designs better than this overestimated one and make further improvement, see Figure 2a. Especially, when the posterior mean of the busy design is extremely small and even exceeds the lower bound of the objective, subsequent optimization can hardly find a better solution. In other words, subsequent optimization will only propose more overestimated designs with even lower predicted performance to "make improvement", even though they are even farther from the ground truth and drive the optimization away from the real meaningful regions. This overestimation issue has not been studied in the past literature to the best of our knowledge, though KB is still the strategy used in popular BO packages (Kandasamy et al., 2020; Balandat et al., 2020b).

**Local Penalization** (LP) (González et al., 2016) is another widely used approach that directly penalizes the nearby region of the busy designs to prevent similar designs from being evaluated next. However, extending this approach to the multi-objective scenario sometimes leads to sub-optimal selection of new designs, as explained in Figure 2b. Intuitively, this sub-optimality comes from the failure of leveraging the accurate predictions from the surrogate model. Consider when selecting the best design to evaluate from a set of candidate designs (A and B) proposed by the multi-objective solver using hypervolume improvement criterion, while a busy design is in evaluation. LP penalizes the nearby regions of the busy design in the design space but has no control over the performance space, which means that designs with similar performance as the busy design could still be selected (design A). Ideally, if the surrogate prediction of the busy design is certain, we can leverage this to avoid proposing designs with little performance gain. For example, simply believing the prediction of the busy design leads to selecting design B that has a higher hypervolume improvement.

### 4.3 BELIEVER-PENALIZER STRATEGY

In conclusion, we observe that the failure case of KB is due to the trust of uncertain predictions while the sub-optimality of LP comes from not believing the certain prediction. Therefore, we propose a novel strategy BP that naturally combines KB and LP by applying KB to designs with certain predictions and LP to designs with uncertain predictions. Here, certainty of prediction is simply defined as the posterior standard deviation from the surrogate model which can be Gaussian processes, Bayesian neural networks or other type of model that computes standard deviation of

predictions. Though the idea of BP is general and one can use any analytical expression to determine the certainty threshold for applying KB or LP, in practice, we find a simple probabilistic form which works well: $P_i(\mathbf{x}) = \max(1 - 2\sigma_i(\mathbf{x}), 0)$ where $P_i$ is the probability of believing $\mathbf{x}$ for the $i$-th objective and $\sigma_i$ is the posterior std of $\mathbf{x}$ from the surrogate model of the $i$-th objective. Because the objective data is normalized as zero with mean unit variance before fitting the surrogate models, $\sigma_i(\mathbf{x})$ is generally between 0 and 1. As a result, BP generalizes more robustly than both KB and LP to most of the benchmark problems as empirically demonstrated in Section 6.1.

## 5 THE AUTOOED PLATFORM

### 5.1 OVERALL WORKFLOW

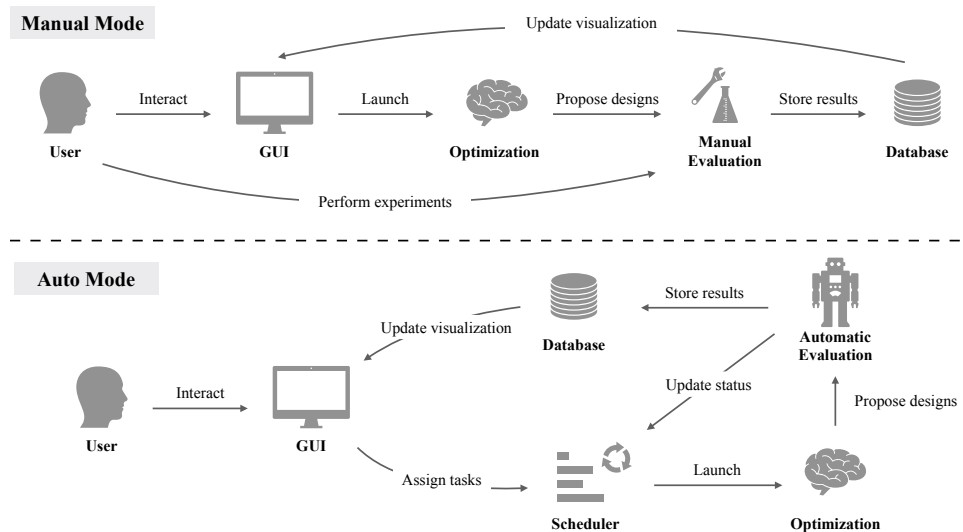

Figure 3: The manual and automatic workflows of the platform design.

As a full-stack software, the overall workflows of AutoOED are presented in Figure 3. The manual mode and auto mode are distinguished by whether the experimental evaluations need to be done manually or can be done automatically through programs. Key components in this workflow include:

- **GUI**: An intuitive graphical interface between users and optimization, evaluation and database, with many powerful functionalities supported as described in Section 5.2.

- **Optimization**: Once the user sends the optimization request, the MOBO algorithm will be running in back-end and proposing next promising designs.

- **Evaluation**: AutoOED supports two different ways of performing evaluations: manually by hand and automatically by evaluation programs. The evaluation module receives designs proposed by the optimization algorithms and outputs the corresponding objective values.

- **Database**: The SQL database stores information of each design including design parameters, predicted and real objective values in a transactional manner which can be distributed.

Next, we illustrate how the workflows of AutoOED combine these individual components.

**Manual mode**   When the experimental evaluation must be performed by hand, the manual mode has to be selected. In this case, users need to interface with both the GUI and evaluation. When GUI receives the optimization request from the user, it launches the optimization worker to propose designs for the user to evaluate. Users will see the new proposed designs from the database GUI, then they can enter the evaluation results in the same interface. Once AutoOED receives the evaluation results, visualizations and statistics will be updated to inform users the latest status.

**Auto mode** If the evaluation program is available (in Python/C/C++/Matlab), AutoOED can automatically guide the experiments by alternating between optimization and evaluation through a scheduler. In this case, upon receiving the optimization request and stopping criteria from the user, the scheduler will automatically repeat the optimization-evaluation cycle until one of the stopping criteria is met. The scheduler automatically launches evaluations after designs are proposed by optimization workers, and restarts optimization after evaluations are done, thus the whole loop of experimental design can be executed in an automated way without human intervention.

## 5.2 FEATURE COMPARISON WITH OTHER PLATFORMS

As shown in Table 1, we compare AutoOED with existing BO platforms according to the following important criteria. For other features of AutoOED, please refer to Appendix A.

**Graphical user interface and visualization** The GUI guides the user through a set of simple steps to configure the problem, such as the number of design and performance parameters, the parameter bounds and constraints, parallelization settings, and selection of the optimization algorithm without the need of coding. The GUI also includes a real-time display of the design and performance space which allows users to easily understand the current status of optimization. We also support displaying and exporting the whole optimization history (including database and statistics). All previous platforms do not offer such a convenient GUI and even the visualizations need to be written by the user, except GPyOpt has a built-in tool for plotting the acquisition function and convergence.

**Multiple objectives and multiple domains** For multi-objectivity, GPflowOpt implements HVPOI (Couckuyt et al., 2014) and Dragonfly implements MOORS (Paria et al., 2020) without the flexibility of incorporating other algorithms or modules. BoTorch supports MESMO (Belakaria et al., 2019), $q$EHVI (Daulton et al., 2020b) and $q$ParEGO. Although algorithm implementations differ across platforms, AutoOED covers a wider range of algorithms and incorporates them into a more unified modular framework. Except continuous designs, AutoOED supports discrete, binary, categorical designs and a mix of them by applying discrete or one-hot transformation in fitting and evaluating the surrogate model, as suggested by Garrido-Merchán & Hernández-Lobato (2020).

**Asynchronous optimization** As demonstrated in Section 4, AutoOED supports different asynchronous techniques including KB, LP and BP, while Dragonfly and BoTorch only implements KB and all other platforms do not support asynchronous optimization.

**External evaluation** There are many real-world experimental design problems where the experimental evaluation must be performed by hand or external lab equipment thus it is hard to write an analytical objective function in code. Though simple to implement, surprisingly, among all of the existing platforms surveyed in this paper, only GPyOpt supports evaluating externally and suggesting designs to evaluate purely based on a given dataset. AutoOED allows users to see the suggested designs and enter the evaluation results easily in the database interface, as described in Section 5.1.

## 6 EXPERIMENTS

We conduct experiments across 12 standard multi-objective benchmark problems: ZDT1-4 (Zitzler et al., 2000), DTLZ1-4 (Deb et al., 2005), OKA1-2 (Okabe et al., 2004) and VLMOP2-3 (Van Veldhuizen & Lamont, 1999). For each algorithm, we run experiments with 20 different random seeds and a budget of 100 samples. The initial 20 samples of each run are generated by Latin hypercube sampling (McKay et al., 1979). We measure as the performance criterion the log of the difference between the hypervolume of the ground-truth Pareto front and hypervolume of the best Pareto front approximation found by the algorithms (thus lower is better). The curves are averaged over 20 random seeds and the variance is shown as shaded regions. Detailed problem information and hyperparameters are described in Appendix B. Additional ablation studies are included in Appendix D.

### 6.1 ASYNCHRONOUS MOBO ALGORITHMS

To test whether Believer-Penalizer is effective, we compare four asynchronous MOBO algorithms on all benchmark problems. *Async* simply ignores the busy designs while optimizing asynchronously

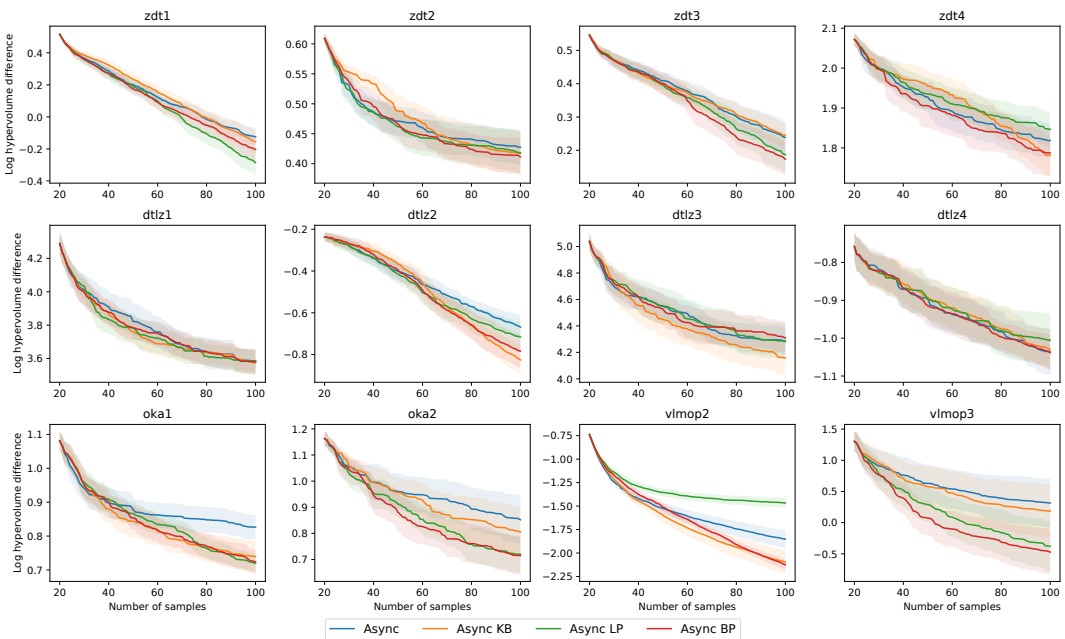

Figure 4: Performance comparison between variants of asynchoronous MOBO algorithms.

and the remaining algorithms are described in Section 4. Figure 4 shows that *Async BP* consistently outperforms other methods and follows the best of *Async KB* and *Async LP*.

## 6.2 PERFORMANCE COMPARISON ACROSS PLATFORMS

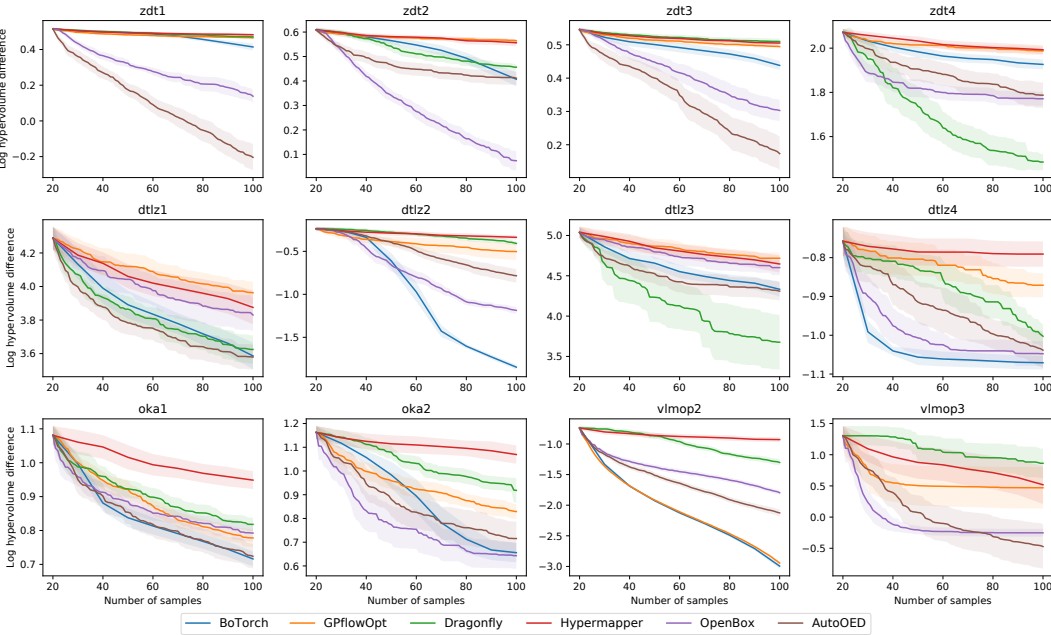

Figure 5: Performance of AutoOED on the benchmark problems compared to other BO platforms. The result of BoTorch on VLMOP3 is omitted as the algorithm fails to stop within 24 hours.

We compare AutoOED against other open-source BO platforms with multi-objective optimization capabilities, including BoTorch, GPflowOpt, Dragonfly, HyperMapper and OpenBox on the afore-mentioned benchmark problems. Our baseline implementation follows the default recommended

settings in the original documentation and tutorials. For BoTorch, we use the $q$EHVI acquisition function. In Figure 5, we demonstrate the competitive performance of AutoOED against other BO platforms using our benchmark. AutoOED takes a major lead in several challenging problems such as ZDT1, ZDT3, and VLMOP3, which shows that our platform handles high-dimensional MOBO problems very well with the proposed asynchronous BP strategy. Besides, the performance of AutoOED is generally stable and ends up either the best or comparable on most benchmark problems. We conduct additional ranked and paired comparisons between AutoOED and all the baseline platforms in Appendix C to further demonstrate AutoOED's robustness and competitive performance.

## 6.3 REAL-WORLD OPTIMAL EXPERIMENTAL DESIGN

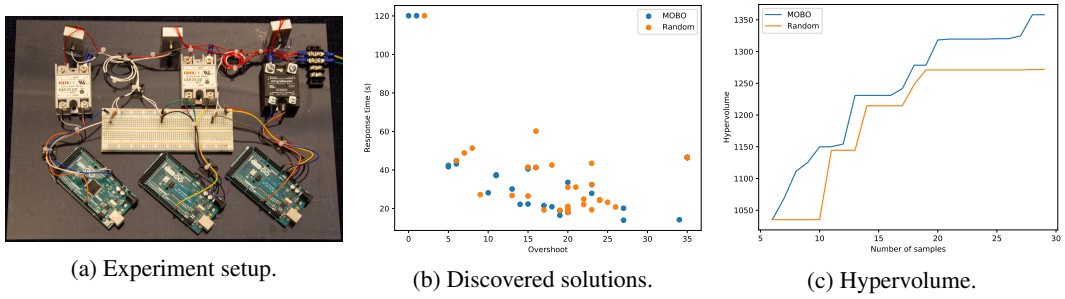

(a) Experiment setup.  (b) Discovered solutions.  (c) Hypervolume.

Figure 6: Physical setup, optimization process and solutions of the PID heater control experiment.

We further demonstrate the real-world applicability of AutoOED through applying to optimize a PID heater controller to have optimal response time and minimal overshoot in a fully automated way. Details of the experimental setup can be found in Appendix B.3. To automate the experiment, we simply link the Python evaluation program of this experiment setup to AutoOED through GUI then start the iterative optimization. Finally, the results are exported as shown in Figure 6, where a set of solutions are discovered with optimal trade-offs between minimal response time and minimal overshoot. Using MOBO algorithms provided by AutoOED is able to discover better designs compared to random sampling at the presence of evaluation noise (temperature measurement error, lack of precise initial temperature control, fabrication differences between heater blocks). This example, with all the components that people can easily buy off-the-shelf, serves as a simple proof of concept that AutoOED is applicable to optimize real physical systems. More examples, including optimizing material structure based on FEM simulation and optimizing a physical motor's rotation, can be found in our documentation with detailed instructions on how to interact with GUI and compose the evaluation program for fully automated OED.

## 7 CONCLUSION AND FUTURE WORK

We introduced an open-source platform for automated optimal experimental design of multi-objective problems. The platform is of modular structure, facilitating the implementation of different multi-objective Bayesian optimization (MOBO) algorithms and enabling both data- and time-efficient optimization. In addition, the platform includes a novel strategy for asynchronous batch optimization for improved time efficiency. We performed extensive experiments on standard benchmark to demonstrate the robust performance against other relevant methods. Furthermore, we conducted real-world physical experiments to showcase the automated usage of our platform.

From the research and engineering perspective, future work includes implementing additional features, such as expensive constraint handling, advanced noise handling and extending AutoOED's framework to incorporate a even wider range of MOBO algorithms. From a practical standpoint, we are also interested in applying AutoOED to automate more real-world experimental design problems in science and engineering. And we believe AutoOED will gradually lower the barrier between MOBO research and practical applications.

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

# A  PLATFORM DETAILS

## A.1  ADDITIONAL FEATURES

**Cross-platform**   AutoOED is a cross-platform software that can be installed as an executable on computers with either Windows/MacOS/Linux system. Installing from source code is also supported for extra flexibility.

**Surrogate prediction**   Users may find AutoOED useful not only in optimization but also in prediction. In addition to the set of optimal solutions, our platform's final product is the learned prediction models of the unknown objectives, which can be used easily from GUI to predict the objectives for a given design from the user. The prediction provides the users with more insights into the potential outcomes of experiments. It helps them better understand the optimization problem to make informed decisions and guide the optimization process towards their preference.

**Constraints**   Besides, AutoOED supports inexpensive black-box constraints on the design space in addition to the bounds. To handle expensive black-box design constraints or objective constraints, it is possible to learn a constraint model similar to learning the objective function, as implemented by Spearmint (Gelbart et al., 2014).

# B  EXPERIMENT SETUP

## B.1  BENCHMARK PROBLEMS

In this section, we briefly introduce the properties of each benchmark problem, including the dimensions of the design space $\mathcal{X} \subset \mathbb{R}^d$ and performance space $\mathbf{f}(\mathcal{X}) \subset \mathbb{R}^m$, and the reference points we use for calculating the hypervolume indicator, which are shown in Table 2. We perform 20 independent test runs with 20 different random seeds for each problem on each algorithm. For each test run of one problem, we use the same initial set of samples for every algorithm, which is generated by Latin hypercube sampling (McKay et al., 1979) using a same random seed. To have a fair comparison, we simply set the reference point $\mathbf{r} \in \mathbb{R}^m$ as a vector containing the maximum value of each objective over the initial set of samples $\{\mathbf{x}_1, ..., \mathbf{x}_k\}$:

$$\mathbf{r} = (\max_{1 \leq i \leq k} f_1(\mathbf{x}_i), ..., \max_{1 \leq i \leq k} f_m(\mathbf{x}_i)).$$

Table 2: Basic descriptions of all the benchmark problems.

| Name | $d$ | $m$ | $\mathbf{r}$ |
|------|-----|-----|------|
| ZDT1 | 30 | 2 | (0.9699, 6.0445) |
| ZDT2 | 30 | 2 | (0.9699, 6.9957) |
| ZDT3 | 30 | 2 | (0.9699, 6.0236) |
| ZDT4 | 10 | 2 | (0.9699, 199.6923) |
| DTLZ1 | 6 | 2 | (360.7570, 343.4563) |
| DTLZ2 | 6 | 2 | (1.7435, 1,6819) |
| DTLZ3 | 6 | 2 | (706.5260, 746.2411) |
| DTLZ4 | 6 | 2 | (1.8111, 0.7776) |
| OKA1 | 2 | 2 | (7.4051, 4.3608) |
| OKA2 | 3 | 2 | (3.1315, 4.6327) |
| VLMOP2 | 6 | 2 | (1.0, 1.0) |
| VLMOP3 | 2 | 3 | (8.1956, 53.2348, 0.1963) |

## B.2  HYPERPARAMETERS

Here we present all the common hyperparameters that AutoOED uses in the experiments.

**Surrogate model** We use the same Gaussian process model as the surrogate for all experiments. We use zero mean function and anisotropic Matern 1/2 kernel, which empirically is numerically stable than popular Matern 5/2 kernel in our experiments. The corresponding hyperparameters are specified in Table 3, which are suggested by TSEMO.

Table 3: GP hyperparameters.

| parameter name | value |
|---|---|
| initial $l$ | $(1, ..., 1) \in \mathbb{R}^d$ |
| $l$ range | $(\sqrt{10^{-3}}, \sqrt{10^3})$ |
| initial $\sigma_f$ | $1$ |
| $\sigma_f$ range | $(\sqrt{10^{-3}}, \sqrt{10^3})$ |
| initial $\sigma_n$ | $10^{-2}$ |
| $\sigma_n$ range | $(e^{-6}, 1)$ |

**Multi-objective evolutionary algorithm** The cheap NSGA-II solver employed in AutoOED's MOBO algorithms by default uses simulated binary crossover (Deb et al., 1995) and polynomial mutation (Deb et al., 1996) for finding the Pareto front of acquisition functions. The initial population is obtained from the best current samples determined by non-dominated sort (Deb et al., 2002). The other hyperparameters are specified in Table 4.

Table 4: NSGA-II hyperparameters.

| parameter name | value |
|---|---|
| population size | 100 |
| number of generations | 200 |
| crossover $\eta_c$ | 15 |
| mutation $\eta_m$ | 20 |

### B.3 REAL-WORLD EXPERIMENT SETUP

In our real-world experiment setup, overall, a PID controller is employed to regulate the temperature of the heater block with proportional, integral, and differential constants. To find the optimal set of constants a number of heating cycles are done asynchronously with the controller using AutoOED. The experiment is comprised of setting the PID constants, then heating the block up to a set temperature, and keeping them at the set duration for 2 minutes. During this time the response time and overshoot are measured. After the heating cycle, the heater is then cooled back down to a starting temperature to prepare for another test with new PID constants.

Specifically, the experimental setup is comprised of 3 heaters with identical dimensions and characteristics. Each heater is comprised of a heater block, heating element, temperature sensor, solid-state relay, power supply, and a controller, shown in Figure 7. To run an experiment, AutoOED sends the PID constants to a controller that is free. Next, the PID controller becomes active and starts regulating the temperature of the heater. The temperature sensor measures the temperature of the heater block. Depending on the constants of the PID controller and the temperautre of the heater block, the controller turns the heating element on or off via the solid-state relay. After a period of 2 minutes where the PID controller is active, the controller stops actively regulating the temperature of the heater allowing the heater to cool. Next, the controller starts to monitor the cooling of the heater via the temperature sensor. It monitors it until the heater block cools to a temperature below a threshold. The amount of time it takes to cool the heater block depends on the temperature that the block was heated to during the active period. Once it sufficiently cools, the controller sends the calculated overshoot and response time to AutoOED and notifies that it is ready to run another experiment.

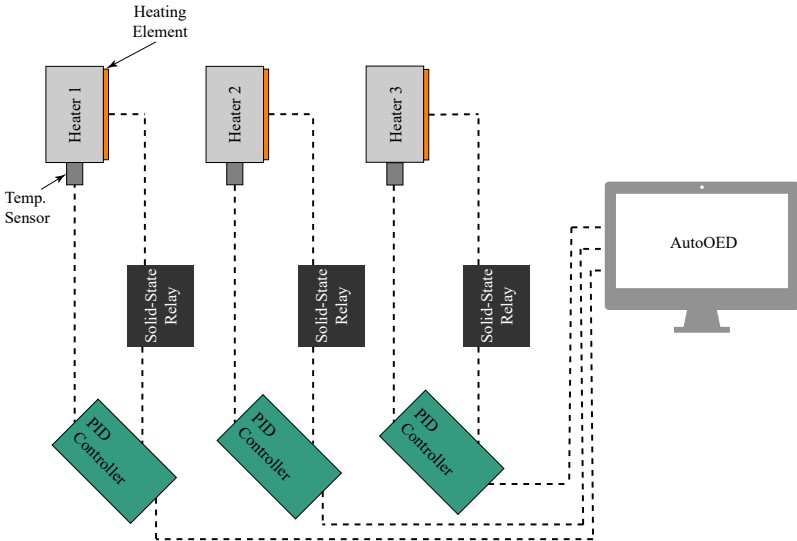

Figure 7: A schematic of the setup used for the real-world experiment.

## C  ADDITIONAL COMPARISONS

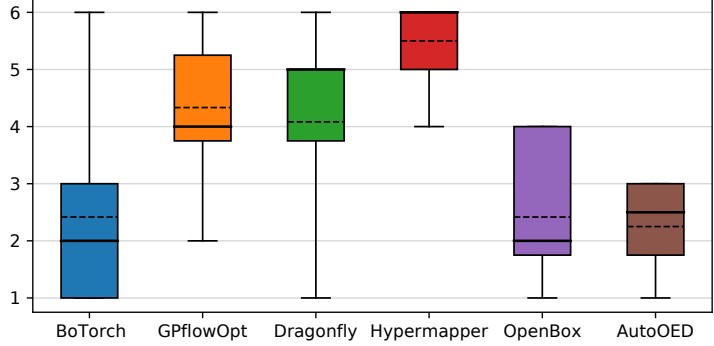

Figure 8: Performance rank of platforms on the 12 benchmark problems (lower is better). The box extends from the lower to the upper quartile values, with a solid line at the median and a dashed line at the mean. The whiskers that extend the box show the range of the data.

We conduct ranked and paired comparisons between AutoOED and all the baseline platforms based on the 12 benchmark problems, as shown in Figure 8 and Figure 9. The performance rank comparison in Figure 8 suggests that BoTorch, OpenBox and AutoOED outperform other platforms by a great margin overall. While BoTorch and OpenBox share a better median rank, AutoOED appears to be the stablest platform that consistently ranks between 1 and 3 on all problems and has a higher lower-bound performance than BoTorch and OpenBox. Figure 9 also suggest that AutoOED has a competitive performance to BoTorch and OpenBox, but outperforms other baselines on a wider range of benchmarks.

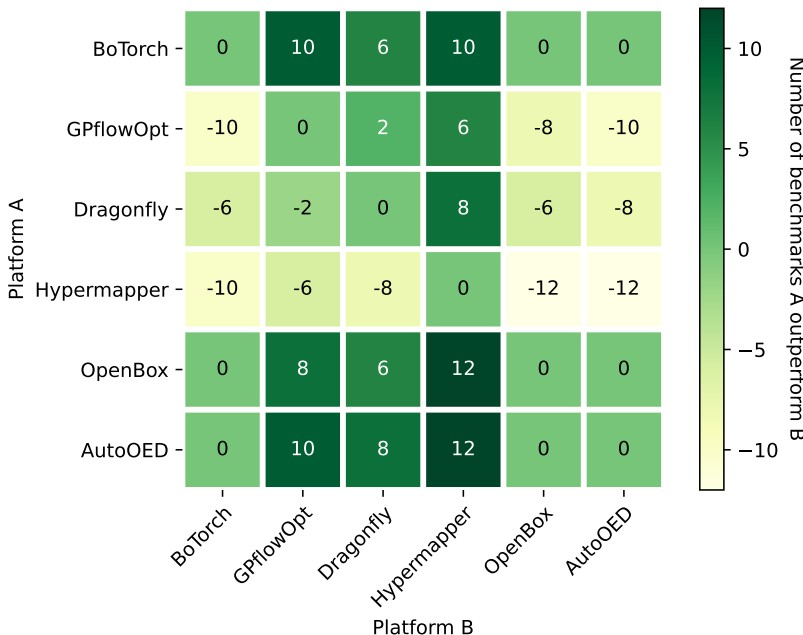

Figure 9: Performance comparison between each pair of platforms on the 12 benchmark problems. Each value in the matrix shows the number of benchmarks that platform A (associated with the row) outperforms platform B (associated with the column), the higher the better.

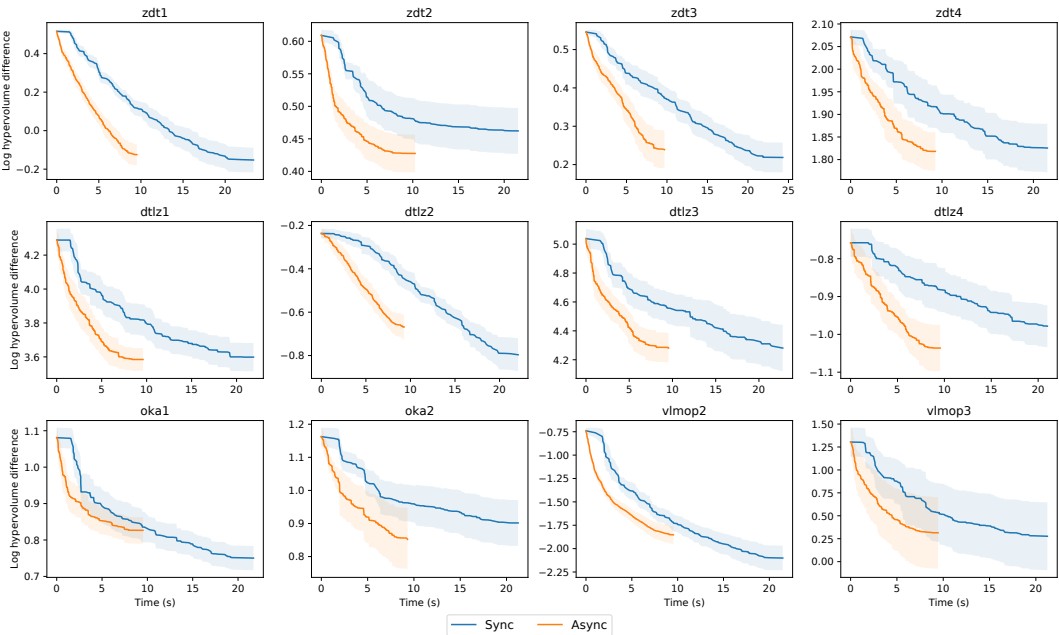

Figure 10: Performance comparison between synchronous and naive asynchronous MOBO algorithms.

# D ABLATION STUDIES

## D.1 SYNCHRONOUS AND ASYNCHRONOUS MOBO

Following the experiment settings in Section 6, we addtionally compare the performance of synchornous and asynchronous MOBO. As shown in Figure 10, they achieve similar hypervolumes whereas asynchronous MOBO spends less than half of the time of its synchronous counterpart.

## D.2 BATCH SIZE

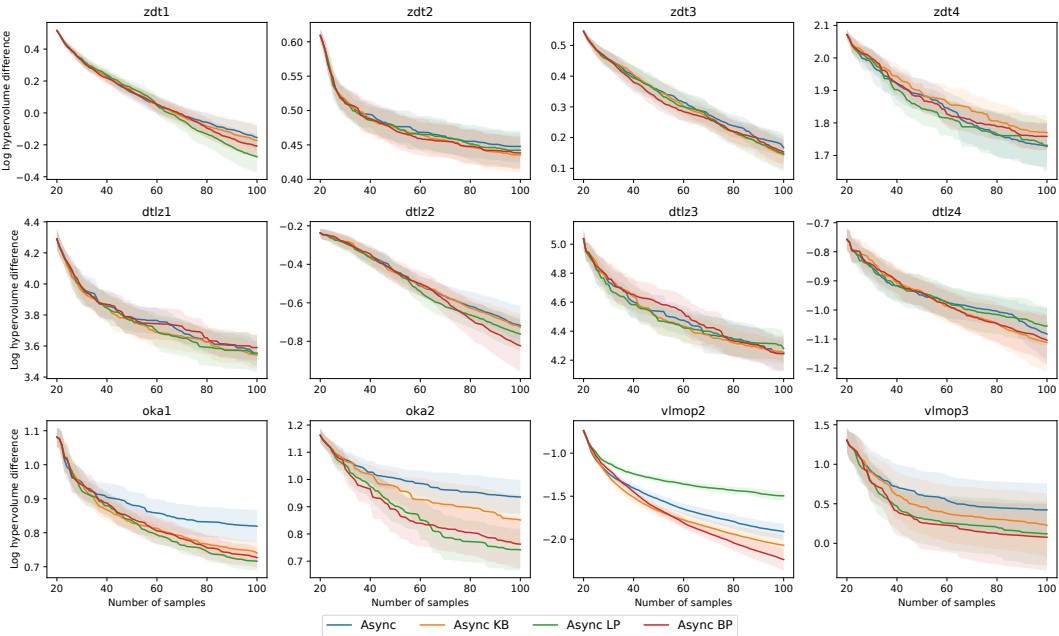

Figure 11: Performance comparison between variants of asynchoronous MOBO algorithms with a batch size of 4.

Ablation studies are also conducted on the batch size in asynchronous MOBO. For this category of experiments, we repeat our practice in Section 6.1 while changing the batch size to 4 and 16, respectively. The results are demonstrated in Figure 11 and 12. Our proposed BP strategy maintains its relative lead in VLMOP3 and performs comparably with other variants on the rest of the test problems.

## D.3 ACQUISITION FUNCTION

Lastly, we evaluate the asynchronous MOBO variants using the EI acquisition function. Although the change in acquisition function has a clear influence on hypervolume growth, the proposed BP variant still performs favorably in problems such as ZDT2, DTLZ2, and VLMOP2. The performance of BP on the other problems remain comparable to the alternative strategies.

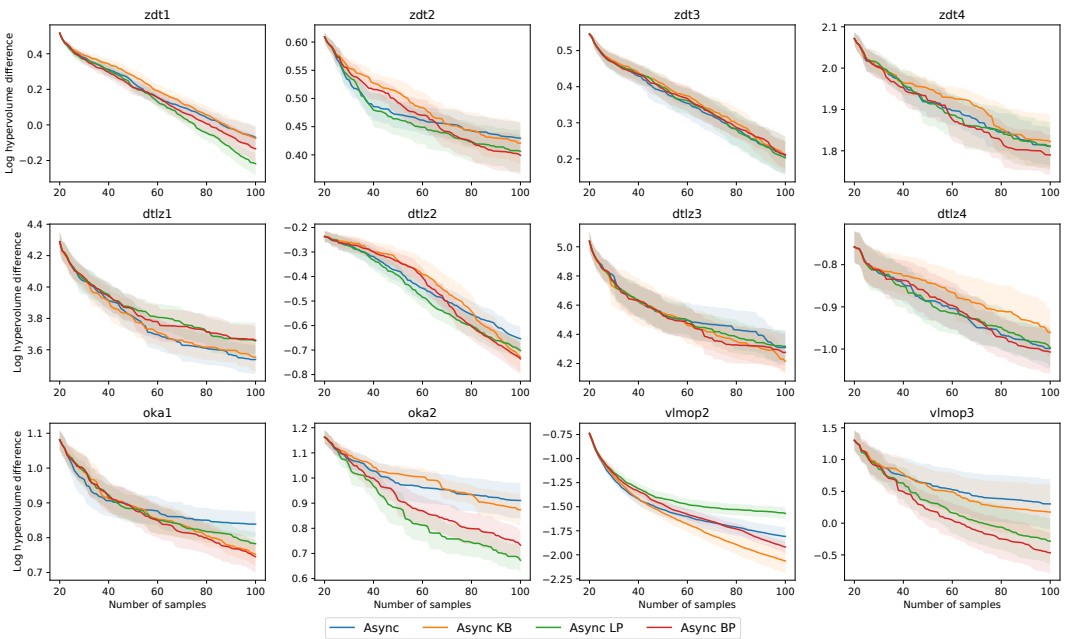

Figure 12: Performance comparison between variants of asynchoronous MOBO algorithms with a batch size of 16.

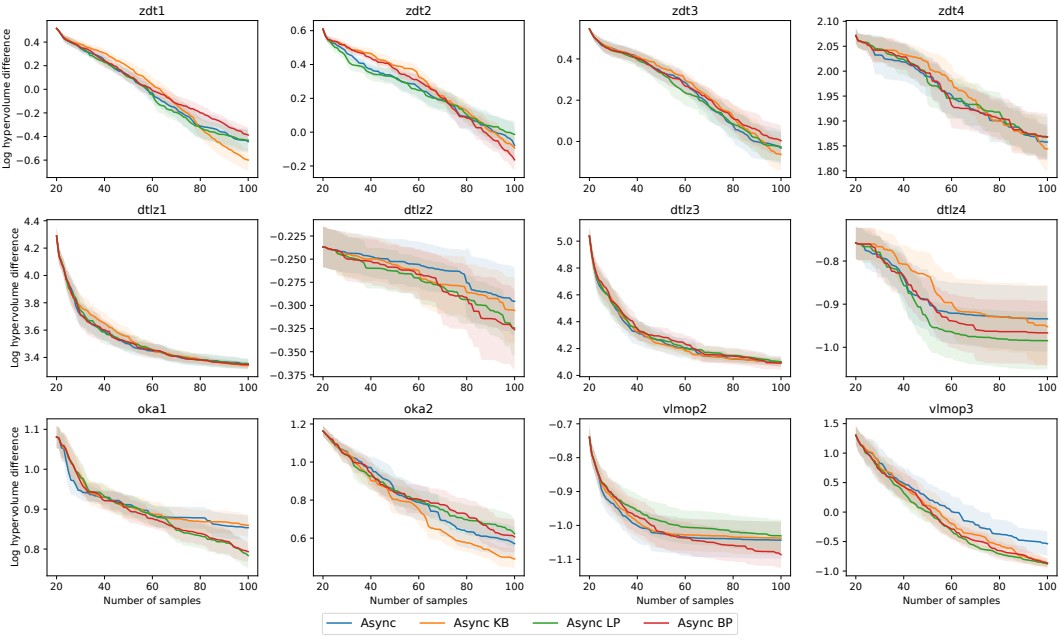

Figure 13: Performance comparison between variants of asynchoronous MOBO algorithms with EI as acquisition function.

