# OpenReview forum: "AutoOED: Automated Optimal Experimental Design Platform with Data- and Time-Efficient Multi-Objective Optimization"
_ICLR.cc/2022/Conference — ICLR 2022 Submitted_

### Official Review · Reviewer_f9tL · 2021-10-21

**Correctness:** 3
**Technical Novelty And Significance:** 3
**Empirical Novelty And Significance:** 3
**Recommendation:** 6
**Confidence:** 3

**Main Review:**

There are several things to like about this paper:

1. The problem studied in this paper is well-motivated. Multi-objective (MO) problems in the context of Optimal Experimental Design (OED) are pervasive. OED in science and engineering often require satisfying several conflicting objectives simultaneously. However, this area of MO-OED has received relatively little attention from the research community.
2.This paper is well-organized and clear written. The authors start from stating the problem formulation. Then, the authors present the algorithm framework for AutoOED.
3. The claims are well supported by four multi-objective tasks on 12 public datasets.

However, I found that there are several shortcomings:

Minor issues:
- in general: better to use Pareto optimal solutions rather than Pareto-optimal solutions
- page 5: I think the authors can directly use BP for Believer-Penalizer as it has been specified in previous section
- in general: I think it is better for the authors to double check about the redundant specifications about the abbreviations throughout the paper
- page 9: importantance should be importance

**Summary Of The Paper:**

In this paper, the authors present AutoOED, an open-source platform for efficiently optimizing multiobjective problems (MO) with a restricted budget of experiments. The platform automatically guides the design of experiments to be evaluated. AutoOED is
built upon multi-objective Bayesian optimization (MOBO).  To accelerate the optimization in a time-efficient manner, the authors propose a strategy called Believer-Penalizer (BP) that allows batch experiments to be accelerated asynchronously without affecting performance. The authors also provide a graphical user interface (GUI) for users to visualize and guide the experiment design intuitively. Finally, we demonstrate that AutoOED can control and guide real-world hardware experiments in a fully automated way without human intervention. Finally, the authors demonstrate that AutoOED
can control and guide real-world hardware experiments in a fully automated way without human intervention.


**Summary Of The Review:**

The problem studied in this paper is well-motivated. Multi-objective (MO) problems in the context of Optimal Experimental Design (OED) are pervasive. I like it very much that the authors would open source such a powerful and convenient framework to complement the current state-of-the-art packages.

---

> ### Author Response · Authors · 2021-11-23
> **Response to Reviewer f9tL**
>
> Thanks for your detailed comments and thoughtful feedback! We encourage you to take a look at our general response first.
>
> We agree that MO-OED is very under-explored so this is our main motivation to design this platform. However, the shortcomings seem to be missing from your review. Would you mind letting us know your suggestions and what can we improve?
>
> Thanks for pointing out the minor issues and we have updated the paper accordingly.

---

### Official Review · Reviewer_QfiR · 2021-10-29

**Correctness:** 3
**Technical Novelty And Significance:** 2
**Empirical Novelty And Significance:** 1
**Recommendation:** 5
**Confidence:** 5

**Main Review:**

### Correctness and Clarity

Overall, the paper is very well written and structured, and easy to follow. Since it is a system paper, the authors focus more on the system parts and describe the advantages and use cases of the system. The only novel contribution from a methodological point of view is Section 4.2 on the Believer-Penalizer strategy for asynchronous parallel optimization. The straightforward and simple idea makes sense; but the threshold for deciding between KB and KP is not well motivated and alternatives are not evaluated.

### Technical Novelty And Significance

The system itself is well designed and implemented. So, from an engineering point of view the system is well justified. However, the system nevertheless gives me the feeling of “yet another Bayesian Optimization package”; over the last years there were so many of them. AutoOED's selling point is the asynchronous, multi-objective optimization which not many systems actually support. However, here AutoOED mostly implements existing techniques and adds with BP only a minor contribution overall.

In the appendix, the authors honestly point out that AutoOED only scales well to 2 or 3 objectives. Although I agree that this is a fairly common use case, there are many applications with more objectives. I encourage the authors to look into the field of many-objective optimization.

### Empirical Novelty And Significance

The empirical results are the weak point of the paper. In Section 6.1, the authors compare BP with KB and LP. They claim that ```BP consistently outperforms other methods```. However, Figure 4 shows that BP is not always the best, sometimes it is the worst performing one (dtlz3). A clear gap between BP and the other baselines cannot be found in any of the benchmarks. I think that a fair statement would be that BP is a robust approach and performs well on average.

The same issue applies to the comparison to other packages. The authors claim ```AutoOED is generally stable and ends up either best or comparable on most benchmarks```. However, Figure 5 shows that AutoOED underperforms on 6 out of 12 benchmarks and is only clearly the best system on 4 benchmarks.

Although I like the benchmark on a real-world benchmark, I have the impression that the benchmark is a bit cooked up and lacks the connection to a real real-world benchmark. Furthermore, AutoOED is only compared against random search on this benchmark.

I appreciate the ablation study in the appendix. However, this gives me the impression that the results for the comparison for KB vs LP vs BP is a bit cherry-picked and it is even more unclear whether BP performs well on other settings of AutoOED.

### Questions to the Authors

1. How does AutoOED perform on single objective benchmarks?
2. You mention “convergence” of AutoOED twice. How is convergence measured or determined?
3. How can a neural network (MLP) be used for BO since it is not a probabilistic model? (see list of surrogate models in Section 3.2)

### Minor Remarks

I strongly disagree with the following statement:

``` However, they are all targeted for experts in coding without an intuitive GUI, which means that the users need to write hundreds lines of code defining the design space, performance space, surrogate models, acquisition functions, optimization solvers, etc. to finally start the optimization.```

In fact, more and more tools get easier to use and most of the listed tools don’t require hundreds of lines; nevertheless some of them do.

In “Open-source Bayesian optimization platform” I’m missing many more open source BO packages, incl. AX, HyperOpt, SMAC, Scikit-optimize, HyperMapper, OpenBox or Optuna. Some of them even support multi-objective optimization. I encourage the authors to extend this list accordingly.


**Summary Of The Paper:**

The paper presents a package for black-box optimization and is specifically designed for the optimization of experimental designs. To this end, the authors build upon multi-objective Bayesian Optimization which allows to obtain good points within a few function evaluations and also enables to obtain a Pareto-Front of non-dominated points. To have an efficient, asynchronous parallelization, the authors propose Believer-Penalizer for choosing the next point while others are still being evaluated. The main idea is to choose either Kriging Believer (KB) and Local Penalization (LP) based on a threshold on the uncertainty in the posterior distribution of the probabilistic surrogate model. In the experiments, the authors show that their package achieves often a fairly good performance.

**Summary Of The Review:**

Overall, AutoOED is a well engineered BO package and it has some strengths and rare features such as multi-objective optimization. However, the novelty is rather minor, there are only little new ideas in the package and the claims in the experiments are clearly overstated.

---

> ### Author Response · Authors · 2021-11-23
> **Response to Reviewer QfiR**
>
> Thanks for your detailed comments and thoughtful feedback! We encourage you to take a look at our general response first. Below we address more specific concerns related to your comments.
>
> First, thank you for acknowledging the engineering contribution of AutoOED. As we summarized in the general response, the main differences between AutoOED and other platforms are the user-friendliness to general users, the modular design that helps machine learning researchers to design and evaluate new modules/algorithms more conveniently, and also the first demonstration of asynchronous MOBO.
>
> Thanks for pointing out that AutoOED was limited by the number of objectives that it can handle. To address this concern, we implemented single-objective and many-objective optimization and visualizations in our platform so now it supports any number of objectives. Thanks to the modular design and the MOBO framework that defines acquisition for each objective independently, we only changed the solver module using existing evolutionary single/multi/many-objective solvers.
>
> For the empirical comparison, we agree that a fair statement should be BP is a robust approach and generally performs well so we updated the paper accordingly. But we disagree that the result is cherry-picked since we have tested it extensively on 12 benchmark problems with ablation studies on both batch size and acquisition functions, which is far more extensive than many previous MOBO algorithm or platform papers. We do not intend to claim BP outperforms all other baselines on all benchmark problems, but we believe it’s fair to claim that BP is generally the most robust approach on all the ablation studies and also AutoOED has competitive performance when compared to other platforms. Besides, we want to further clarify that the real-world example is a proof of concept rather than a benchmark -- we show that AutoOED can be easily integrated into the optimization of physical devices that people can buy off-the-shelf.
>
> To answer your questions:
>
> Q: How does AutoOED perform on single objective benchmarks?
>
> A: AutoOED is mainly optimized and targeted for the multi-objective setting since MOBO software is much less invented. We include single-objective BO supports as well but have not benchmarked them.
>
> Q: You mention “convergence” of AutoOED twice. How is convergence measured or determined?
>
> A: We actually mention “convergence” once for GPyOpt and once for AutoOED. For GPyOpt, plotting the convergence simply means plotting the best performance w.r.t. optimization iterations. For AutoOED, we are referring to the convergence of hypervolume, which is a stopping criterion that can be set by the user and is determined by if the hypervolume stops to improve over a certain number of past iterations.
>
> Q: How can a neural network (MLP) be used for BO since it is not a probabilistic model? (see list of surrogate models in Section 3.2)
>
> A: MLP is only compatible with the identity acquisition function which does not require uncertainty estimation. For example, DGEMO (Konakovic Lukovic et al. 2020) is a successful MOBO algorithm that does not require the surrogate model to be probabilistic. This sounds counter-intuitive and technically we don’t know if this kind of approach can still be classified as BO, but this is empirically feasible and we observed that for multi-objective BO, uncertainty estimation is sometimes not as much necessary as that in single-objective BO.
>
> Finally, we agree that more and more tools are becoming easier to use and we are glad to see that trend. We have revised that statement in the paper. But as we discussed in the response to all reviewers, it is still hard for people without coding skills to set up the environment, install packages and learn Python and APIs (perhaps much harder than writing actual codes). The major advantage of AutoOED is the easy installation and the whole OED process can be fully done with GUI interactions. Therefore, our goal aligns with those packages, which is to lower the barrier for people to use MOBO techniques. In addition, thanks for suggesting more relevant BO packages to us and we have updated the paper with the quantitative comparisons to them.

---

> > ### Comment · Reviewer_QfiR · 2021-11-29
> > **Thanks for the Reply**
> >
> > Thank you very much for the thorough reply.
> >
> > Overall, I really see the benefit of AutoOED and I thank the authors for implementing it. I believe that this tool is a great step forward from an implementation and usability point of view.
> >
> > Nevertheless, I'm still not convinced by the overall performance of AutoOED. Figure 5 shows a fairly mixed picture of its performance. Furthermore, there is only little novelty in the implemented method and even that is a fairly straightforward idea.
> >
> > Overall, I increased my score to 5, but am still more on the side of rejection.

---

### Official Review · Reviewer_iLNv · 2021-11-01

**Correctness:** 3
**Technical Novelty And Significance:** 2
**Empirical Novelty And Significance:** 2
**Recommendation:** 5
**Confidence:** 5

**Main Review:**

The authors have designed a generalized platform for further users to quickly design and evaluate their own MOBO algorithms, i.e., the proposed MOBO strategies are applied for data-efficient experimentation; The proposed Believer-Penalizer (BP) optimization strategy is introduced for time-efficient experimentation. The authors organize their work in a well-reading manner. However, the main concern of this paper is that not sure whether it has enough scientific contribution, for instance, the authors claim that they proposed the state-of-the-art strategy named BP while in the paper it is only presented in a very naive way: a novel strategy Believer-Penalizer (BP) that naturally combines KB and LP by applying KB to designs with certain predictions and LP to designs with uncertain predictions. To be honest, I do not have very clue that how authors combine these two together, and it lacks clear proof that in theory how this kind of combination could work, simply experiment results can not be adopted as evidence.

**Summary Of The Paper:**

The authors present an Automated Optimal Experimental Design platform aiming at accelerating discovering solutions with multi-objective problems in a data-efficient manner. The main contribution of the work is introducing a novel strategy for batch experiments acceleration.


**Summary Of The Review:**

this paper is more engineering-oriented, lacking convincing theoretical works.

---

> ### Author Response · Authors · 2021-11-23
> **Response to Reviewer iLNv**
>
> Thanks for your detailed comments and thoughtful feedback! We encourage you to take a look at our general response first. Below we address more specific concerns related to your comments.
>
> Though we agree that this paper is more system-oriented, we have technical contributions from both engineering and scientific perspectives (summarized in our reply to all reviewers). Also, ICLR certainly considers software platforms as one of the related subject areas so we believe engineering contributions should be equally valued. We look forward to enriching the toolbox of AutoOED and making it more convenient for industry-level usage.
>
> For scientific contributions, it is worth noting that AutoOED is the first platform that introduces asynchronous MOBO and we have done the first empirical evaluations on asynchronous MOBO methods. The motivation for developing BP is mainly because we found the failure cases of both KB and LP as we described in the paper, thus we propose a natural and simple form of combining those together, which also addresses the failure cases as we discussed to some extent. But we agree a theoretical proof is also important, and a fairer statement for BP would be it’s a generally more robust strategy rather than a state-of-the-art one. We have updated the paper and added more explanations on this.

---

### Official Review · Reviewer_Bmro · 2021-11-02

**Correctness:** 3
**Technical Novelty And Significance:** 3
**Empirical Novelty And Significance:** 3
**Recommendation:** 5
**Confidence:** 4

**Main Review:**

My main concern about this work is that it is more like an assembly of software design docs and a manual for an open-source platform. There is very limited methodology contribution from this work. The main contribution of this work is the highly modularized design, compared with existing Bayesian optimization platforms, e.g., Spearmint, Dragonfly, etc.

Another issue is that the authors claim at the beginning that ‘The goal is to simultaneously minimize m>=2 objective functions’ but later explain that ‘AutoOED is designed for cases when the number of objectives equals 2 or 3’.  It is misleading to present the platform in this way and might confuse readers with the impression that this is a generic platform to solve multi-objective optimization. Also, it is strange that AutoOED does not apply for only one objective, which should be naturally covered by an MOO algorithm.

I am also surprised that the authors did not cite the following work though it is only on arXiv but strongly correlated with this manuscript.

Tian, Yunsheng, et al. "AutoOED: Automated Optimal Experiment Design Platform." arXiv preprint arXiv:2104.05959 (2021).


**Summary Of The Paper:**

This paper proposed and explained an open-sourced experimental design platform powered using Bayesian multi-objective optimization to accelerate discovering solutions with optimal objective trade-offs. The authors present how to use this platform in detail.

**Summary Of The Review:**

Overall, this is a decent document and manual for MOO software package/platform introduction but not a scientifically methodological paper for ICLR.

---

> ### Author Response · Authors · 2021-11-23
> **Response to Reviewer Bmro**
>
> Thanks for your detailed comments and thoughtful feedback! We encourage you to take a look at our general response first. Below we address more specific concerns related to your comments.
>
> Though we agree that this paper is more system-oriented, we have technical contributions from both engineering and scientific perspectives (summarized in our response to all reviewers). Also, ICLR certainly considers software platforms as one of the related subject areas, so we believe engineering contributions should be equally valued. We look forward to enriching the toolbox of AutoOED and making it more convenient for industry-level usage. In addition, we would like to clarify that this paper focuses on illustrating the design principles, functionalities, uniqueness, and performance of AutoOED, rather than the user manual or detailed documentation, which you can also find on our website.
>
> Thanks for pointing out that AutoOED was limited by the number of objectives it can handle. To address this concern, we implemented single-objective and many-objective optimization in our platform, so now it supports any number of objectives. Thanks to the modular design and the MOBO framework that defines acquisition for each objective independently, we only changed the solver module using existing evolutionary single/multi/many-objective solvers.

---

### Public Comment · ~Susmit_Jha1 · 2021-11-11
**Nice piece of work; a few requests for some clarifications w.r.t BoTorch**

This is a nice piece of work, and the AutoOED tool could prove to be a framework of choice for Bayesian Optimization beyond existing tools such as BoTorch or GPflowOpt.

 Looking at table 1 where AutoOED is compared with other Bayesian tools, it appears it has a few advantages over BoTorch:
- GUI
- Multiple Domains
- External Evaluation
- Built-in Visualization

Leaving aside the GUI/Visualization features (which are great to have, but keeping focus on new technical capabilities), it appears AutoOED can support multiple domains and external evaluation beyond what BoTorch can support.

*Multiple domains*: Elsewhere in the paper, when describing multiple domains, the authors describe that "BoTorch supports MESMO (Belakaria et al., 2019), qEHVI (Daulton et al., 2020b) and qParEGO. Although algorithm implementations differ across platforms, AutoOED covers a wider range of algorithms and incorporates them into a more unified modular framework." It would be helpful to list the additional domains/optimization implementations that are available in AutoOED beyond BoTorch. It would help practitioners easily recognize when AutoOED would be more suitable to use than BoTorch.

*External evaluation* : The aspect will also benefit from further elaboration. The paper mentions "real-world experimental design problems where the experimental evaluation must be performed by hand or external lab equipment thus it is hard to write an analytical objective function in code".... "AutoOED allows users to see the suggested designs and enter the evaluation results easily in the database interface". Is the intent here to support external oracles (beyond analytical functions) or to have human-in-the-loop optimization where humans populate the value for some evaluations?  BoTorch allows external function evaluation - for example, the Hartmann function from the test_functions used in the documentation here: https://botorch.org/tutorials/closed_loop_botorch_only . Having human enter the value appears to be a user-interface challenge. Some additional discussion on this will help readers better appreciate the value of "External evaluation" and understand how it improves on existing Bayesian optimization frameworks.

Also, BoTorch has an AX wrapper that is supposed to improve its usability and address some shortcomings w.r.t user interface: https://botorch.org/docs/botorch_and_ax

As this comment would reflect, it is coming from a practitioner who is using BO tools and is trying to ascertain the incremental gain of using AutoOED versus BoTorch. If the authors feel that there would be quite a few folks with similar questions, it would be great to have discussions on these questions added to the paper or some documentation of the tool.

---

> ### Author Response · Authors · 2021-11-29
> **Advantages of AutoOED over BoTorch**
>
> Thank you for your thoughtful comments and suggestions!
>
> Before clarifications of other technical capabilities, we would like to first emphasize that the GUI/visualization features are extremely important especially for general users (which are the majority of people who need OED), which is the primary motivation for us to develop AutoOED. For other advantages of AutoOED over BoTorch:
>
> **Multiple domains**
>
> We want to clarify that here domain essentially means the data type of design variables. AutoOED supports optimizing continuous, discrete, binary, categorical, and mixed types of design variables, while to the best of our knowledge, BoTorch only supports optimizing continuous variables.
>
> **Additional optimization implementations**
>
> This is actually more related to your questions on multiple domains. BoTorch, along with other existing MOBO platforms, only implements the MOBO framework in a way that is more like single-objective optimization: the acquisition function is to compute a single objective, either entropy-based (MESMO) or hypervolume-based (EHVI). In this framework, a single-objective solver is applied to solve for the candidate designs to evaluate.
>
> in AutoOED, we support a different yet more general MOBO framework -- besides supporting the kind of acquisition function that computes a single objective like in BoTorch, we support defining separate acquisition functions for each objective (e.g. EI, UCB), then applying a multi-objective solver (e.g. NSGA-II) to solve for the candidate designs and finally using a selection scheme (e.g. HVI) to select the batch of designs to evaluate. Although there is no consensus that which framework is more effective, empirically we found they are performing comparably well. The algorithms implemented in our framework are much faster with a larger number of objectives (e.g. DGEMO, TSEMO) while BoTorch will be very slow for a number of objectives >= 3 possibly due to the Monte-Carlo acquisition. Also, many-objective solvers (e.g., NSGA-III) can be applied in our framework for more efficiently exploring the high-dimensional space.
>
> **External evaluation**
>
> For external evaluation, we mean human-in-the-loop optimization where humans populate the value for some evaluations. We agree this is more like a user-interface challenge, but we think user-interface challenges like this are essentially where software platforms become important. There is nothing that people cannot do with coding from scratch, but we don't want every user of our platform to waste time on this.
>
> **Asynchronous optimization**
>
> BoTorch supports asynchronous optimization only in Kriging-Believer fashion, while AutoOED supports Local Penalization and Believer-Penalizer methods which are empirically proven to be more effective in general.

---

### Author Response · Authors · 2021-11-23
**Response to all reviewers**

We thank all the reviewers for their thoughtful suggestions for improving this platform. The paper is updated accordingly with your valuable feedback. Overall, we would like to clarify our aim and contributions, respond to common concerns, and discuss the future works that can be inspired by our platform.

**Motivation and target audience**

Most scientists and engineers nowadays still rely on the manual or heuristics-based design of experiments. Even though MOBO has been extensively studied but not often employed because they are not comfortable with coding, setting up a coding environment, working with the command line, or learning APIs. In addition to general users, machine learning researchers can also benefit from our platform because our modular framework can accelerate designing and iterating new algorithms.

**Engineering contributions**

Easy-to-use: Unlike the complexity of playing around with existing BO packages as we mentioned above, for AutoOED, people can simply download the executable file from our website and run it without configuring the environment and learning coding. The whole OED process can be done with simple GUI interactions and people can see the progress in real-time.

Modular design: With multiple choices existing for each module, we are able to combine and construct new algorithms modularly with a few lines of code, resulting in a much larger space for exploring good algorithms. Thanks to this, we have surprisingly found a new algorithm that is quite simple and empirically performs well (Gaussian Process + UCB + NSGA-II + HVI) but this combination is not mentioned in any previous literature.

**Scientific contributions**

Asynchronous MOBO: It is worth noting that AutoOED is the first platform that introduces asynchronous MOBO and we have done the first empirical evaluations on this, which is missing from the existing literature and software.

Believer-Penalizer: KB and LP are originally from the single-objective setting but we extended it to the multi-objective setting. Even though there are standard methods, we point out their failure cases that, to the best of our knowledge, were not presented in previous literature (see Section 4.2). We found that KB and LP can complement each other to some extent and this observation inspired our BP method.

Evaluations: We have done extensive evaluations on 12 benchmark problems using 20 random seeds with several ablation studies. It is important to clarify that we are not aiming at outperforming all the existing platforms in terms of algorithmic performance. However, we guarantee a competitive and robust performance (for both BP and AutoOED) overall based on our extensive evaluations while being much more easy-to-use and modular. We rephrased our words in describing the results in our paper as suggested by the reviewers to avoid confusion.

**Addressing reviewers’ common concerns**

Lack of scientific contribution and mostly engineering work: We submitted this work to ICLR because software platforms and parallelization are subject areas that ICLR considers: https://iclr.cc/Conferences/2022/CallForPapers. Therefore, we hope our insights from the system perspective will be equally valued as algorithmic innovations. Furthermore, we hope our work would be of interest to other disciplines (such as chemistry, biology, mechanical engineering, etc.) and communities and would bring ICLR to their attention.

AutoOED only works for 2 or 3 objectives: We agree that this was a major limitation so we incorporated optimization methods and visualizations for both single-objective and many-objective scenarios in AutoOED and now it supports any number of objectives.

The theoretical insight behind BP: Although we are unable to give theoretical proof since KB is also not investigated theoretically, we define an intuitive formulation of BP and show its empirically robust performance. We revised the paper with a more intuitive explanation and analysis of BP.

More comparison: As Reviewer QfiR suggested, we cited more existing BO packages and also conducted a quantitative comparison with them on multi-objective benchmark problems (see Figure 5, 8, 9) and AutoOED still maintained competitive performance.

**Inspiration for future work**

We believe that AutoOED can inspire future work in more practical OED applications, such as accelerating the discovery of high-performance materials and chemicals, or any process optimization applications. As an easy-to-use platform, AutoOED aims at bridging the gap between research and practical application by lowering the barrier of machine learning-based OED. Another exciting future work enabled by a modular platform is the adaptive selection of module combinations given the current optimization status (e.g., adaptive exploration-exploitation trade-off). Finally, asynchronous MOBO has not been explored at all and we hope the insights from this paper can incentivize more future work on this topic.

---

### Author Response · Authors · 2021-11-29
**Updated project website and performance comparison**

Dear Reviewers,

Thank you for your effort in reviewing this paper. We appreciate your time in reading our response and additionally, here we share more updated information about our platform:

We have updated our website (https://sites.google.com/view/autooed) with a video introduction and download links for executable files. Hope the updated materials illustrate better that AutoOED is an easy-to-use platform for general users.

Besides, in case we have not made this clear, we updated the performance comparison with more platforms (see Figures 5, 8, 9 in the paper) and hope it clarifies that AutoOED has competitive and also robust performance.

We will continuously maintain and improve AutoOED to be an ideal platform for OED with both ease of use and robust performance. Please let us know how we can better address your concerns and we are happy to answer any remaining questions. Thank you.

---

### Decision · Program_Chairs · 2022-01-20

**Decision:**

Reject

**Comment:**

The reviews are of good quality. The responses by the authors are commendable, but reviewers remain of the opinion that the scientific contribution of the paper is limited, no matter how strong the software engineering contribution may be.